# Effect of Respite Time before Live Transportation on Muscle Quality of Blunt Snout (Wuchang) Bream

**DOI:** 10.3390/foods11152254

**Published:** 2022-07-28

**Authors:** Ling Peng, Juan You, Lan Wang, Shanbai Xiong, Qilin Huang, Tao Yin

**Affiliations:** 1College of Food Science and Technology, National R&D Branch Center for Conventional Freshwater Fish Processing (Wuhan), Huazhong Agricultural University, Wuhan 430070, China; pengling@webmail.hzau.edu.cn (L.P.); juanyou@mail.hzau.edu.cn (J.Y.); xiongsb@mail.hzau.edu.cn (S.X.); hql@mail.hzau.edu.cn (Q.H.); 2Institute of Agricultural Products Processing and Nuclear-Agricultural Technology, Hubei Academy of Agricultural Sciences, Wuhan 430064, China; 2005lily@gmail.com

**Keywords:** blunt snout bream, respite time, live transportation, texture, cell structure

## Abstract

To provide scientific support for improving the muscle quality of blunt snout bream, ultrahigh performance liquid chromatography, texture analyzer, and optical electron microscopy were applied to explore the effects of respite time (0, 24, 48, and 72 h) on the muscle quality of blunt snout bream before live transportation. The energy compounds (ATP and glycogen) of muscle significantly decreased with the respite time (*p* < 0.05). Lactic acid content declined and then increased, leading to a rise and then a fall in pH (*p* < 0.05). Water-holding capacity of fish muscle increased progressively when the respite time was increased to 48 h and then dramatically decreased when the respite time was further increased to 72 h (*p* < 0.05). Shear force gradually increased (*p* < 0.05), while the whiteness and lightness values remained stable (*p* > 0.05). Both the content of umami compounds (IMP) and bitter compounds (HxR, Hx), and the calculated K value decreased steadily with the respite time (*p* < 0.05). The extracellular gap of the fish muscles gradually decreased with the respite time. The most uniform and intact cellular structure occurred at 48 h. However, when the respite time was extended to 72 h, the extracellular gap and muscle fragmentation rate of the muscle increased considerably. The findings indicated that a 48 h respite time was suitable to improve the muscle quality of blunt snout bream after live transportation.

## 1. Introduction

At present, cross-regional logistics of freshwater fish is mostly undertaken using live fish transportation vehicles equipped with oxygen and water tanks, to resolve the issue of unbalanced supply and demand for fishing resources [1]. The live freshwater fish is transported in closed containers with a restricted amount of water; thus, mucus generated by fish and suspended particles in feces quickly contaminates water bodies [2]. Additionally, ammonia (a compound of fish metabolism) is dissolved after being expelled via the fish’s gills. When the quantity of ammonia nitrogen exceeds 2 mg/L in the water, fish may die [3]. Respite before live transportation of fish is one of the most practical and effective methods for reducing transportation stress and increasing survival rates [4]. During the respite, the fish expel their excreta, thereby minimizing ammonia nitrogen and odorous chemical buildup in the culture environment [5,6]. This will also reduce metabolism rate and oxygen consumption during live transportation, lessen stress, and ultimately boost survival time and rate [7]. Domestic and international research on respite treatment has focused on its influence on ammonia nitrogen metabolism and survival time of the fish during the live transportation [8,9,10].

The fish’s body is divided into four parts: the head, torso, tail, and fins, with the muscle of the torso being the major part used as food [11]. Muscle quality affects customer’s acceptance, which ultimately determines economic value. Muscle quality (shear force, water-holding capacity, color, pH of the muscle, etc.) is affected by the fish stress [12,13,14]. Yu et al. [15] discovered that feeding grass carp (*Ctenopharyngodon idella*) only by faba bean (*Vicia faba L.*) enhanced hardness and elasticity of muscle greatly, which was due to the diet stress. Acerete et al. [16] demonstrated that pre-slaughter anesthetics and exposure to low temperatures reduced stress levels, which decreased lactic acid buildup in fish muscle, and enhanced sensory (odor, gill, skin, and eye) scores in European perch (*Dicentrarchus labrax*). Lv et al. [17] reported that purification treatment could improve muscle hardness, flexibility, and water-holding capacity (WHC) of the grass carp muscle, along with decrease the odoriferous volatile chemicals (nonanal and hexanal). Respite treatment has been reported to reduce transportation stress. However, no studies have been conducted on the impact of respite treatment on the muscle quality of fish after live transportation.

Blunt snout bream (*Megalobrama amblycephala*) is a type of bream widely farmed in China. It is popular with customers due to its high protein and vitamin content and low-fat content. Therefore, this study simulated the commercial transportation of live blunt snout bream (short-time respite, capture, and live transportation). Additionally, ultrahigh performance liquid chromatography, texture analyzer, and optical electron microscopy were applied to investigate the effects of respite time on the muscle quality of blunt snout bream after live transportation, in order to provide scientific support for improving the muscle quality of blunt snout bream.

## 2. Materials and Methods

### 2.1. Materials

Blunt snout bream with an average length of 35.1 ± 1.39 cm and weight of 641.25 ± 56.58 g was obtained from a pond in Ezhou National Original Breeding Farm (Ezhou, China). The water temperature in this pond was 16.68 °C, the dissolved oxygen was 12.49 mg/L, and the pH was 8.14.

Anhydrous ethanol, sodium hydroxide, phosphoric acid, potassium dihydrogen phosphate, and HE dye suit were purchased from Sinopharm Chemical Reagent Co., Ltd. (Shanghai, China). Acetonitrile and methanol were purchased from Merck (Darmstadt, Germany).

### 2.2. Sample Preparation

All animal standard operation procedures were approved by the Animal Care and Use Committee of Huazhong Agricultural University and performed in accordance with the Guidelines for Care and Use of Laboratory Animals of Huazhong Agricultural University. Blunt snout bream of uniform size and health status were respited (0, 24, 48, 72 h) in a net box (6 m × 1 m × 1 m) at 3 m offshore, with the bottom of the net box 0.5 m from the lake bottom. At the end of each time period of respite, blunt snout bream (10 samples) was captured from the net box and transferred to a plastic box (54.5 cm × 37 cm × 34.5 cm) with a doubled amount of water (water from the same pond) according to their weight, and ice bags were added. The live blunt snout bream were transported to the laboratory within 2 h, and then the boxes were transferred to a simulated transport platform (DK-5024, Starshow Intelligent Equipment Co., Ltd., Xiamen, China) for further transportation under a vibration frequency of 120 r/min. During the whole transportation process, the water temperature was maintained at 10 ± 3 °C and dissolved oxygen at >10 mg/L. The transportation procedure and conditions adopted in the experiment were referenced to the commercialization model of transporting blunt snout bream in China [1].

### 2.3. Lactic Acid and Muscle Glycogen

Lactic acid and muscle glycogen content were determined by kits (A019-2-1/A043-1-1, Nanjing Jiancheng Institute of Biological Engineering, Nanjing, China). At 620 nm, glycogen was detected, whereas lactic acid was detected at 530 nm. Three samples were used in the determination for each treatment.

### 2.4. pH

The electrode of an insertion pH meter (TESTO 205, DETO Instruments International Trading Co., Ltd., Shanghai, China) was inserted into the fish muscle, and the pH value of the dorsal muscle of the blunt snout bream was shown on the display. Six samples were used in the determination for each treatment.

### 2.5. Water-Holding Capacity

Water-holding capacity was measured according to the method of Subbaiah [18] with slight changes. About 3 g of fish muscle was weighed and wrapped in a double layer of qualitative filter paper and centrifuged at 4000 rpm for 15 min. The water-holding capacity was expressed as the ratio of the sample mass before and after centrifugation. Six samples were used in the determination for each treatment.

### 2.6. Color

Color was measured according to the method of Shi et al. [19] with slight changes. The dorsal muscle of blunt snout bream was cut into cubes (20 mm × 20 mm × 10 mm), and *L**, *a**, and *b** values were recorded using a portable colorimeter (CR-400, Konica Minolta, Tokyo, Japan). Muscle samples without respite (0 h) were used as a reference for Δ*E* calculation. Six samples were used in the determination for each treatment.
(1)$$W=100-\sqrt{{\left(100-L*\right)}^{2}+{a*}^{2}+{b*}^{2}}$$
(2)$$△E=\sqrt{△{L*}^{2}+△{a*}^{2}+△{b*}^{2}}$$
where *W* indicates the whiteness of the sample; Δ*E* indicates the total color difference; *L** value indicates the lightness of the sample: positive *a** represents red, negative *a** represents green, positive *b** represents yellow, and negative *b** represents blue (based on AMSA Meat Color Measurement Guidelines, 2012).

### 2.7. Shear Force

Shear force was measured according to the method of Shi et al. [19] with slight changes. The dorsal muscle of blunt snout bream was cut into 20 mm × 20 mm × 10 mm cubes. Cubes were cut perpendicular to the direction of muscle fibers at a speed of 60 mm/min using a texture analyzer (SD-700, Akiyama Technology Co., Ltd., Dongguan, China) equipped with a blade (15 mm in diameter). After cutting, the shear force was recorded as the maximum force (g). Ten samples were used in the determination for each treatment.

### 2.8. ATP-Related Compounds

Separation and identification of samples were performed using ultrahigh performance liquid chromatography (Acquity UPLC-H Class, Waters Corporation, Milford, MA, USA), according to Liu [20]. The separation was performed on an ACQUITY UPLC BEH Amide (1.7 μm, 2.1 mm × 100 mm, Waters) column. The mobile phases were acetonitrile, 10 mmol/L sodium dihydrogen phosphate, and 0.1% (*v*/*v*) aqueous phosphate solution; the gradient system consisted of varying concentrations of mobile phases A, B, and C (0–6 min, 88–80% A, 7–17.5% B; 6–8 min, 80–77% A, 17.50–22% B; 8–9 min, 77–65% A, 22–35% B; 9–10.7 min, 65–55% A, 35–45% B; 10.70–10.80 min, 55–88% A, 45–7% B; 10.80–23.00 min, 88% A, 7% B). The chromatographic separation conditions were set as follows: column temperature 50 °C; flow rate 0.5 mL/min; injection volume 5 μL. Based on ATP-related compounds, the K value is usually calculated as the percentage rate of HxR and Hx to the sum of ATP and degradation products. Three samples were used in the determination for each treatment.
(3)$$K=\frac{\mathrm{Hx}+\mathrm{HxR}}{\mathrm{ATP}+\mathrm{ADP}+\mathrm{AMP}+\mathrm{IMP}+\mathrm{Hx}+\mathrm{HxR}}\times 100\%$$

### 2.9. Morphological Observation

Morphological observation was carried out according to the method of Shi et al. [19] with slight modification. The dorsal muscle was fixed with 4% paraformaldehyde solution overnight. The fixed sample was sliced in paraffin, dewaxed, stained with hematoxylin eosin, dehydrated, and sealed. The processed samples were scanned panoramically with an optical microscope (EclipseCi, Nikon, Tokyo, Japan) and displayed using a Pannoramic Viewer (1.15.3, 3DHISTECH Ltd., Budapest, Hungary). The magnification of scanned images was adjusted to 100 times. Three samples were used in the determination for each treatment.

### 2.10. Statistical Analysis

SAS software (V8, SAS Institute Inc., Cary, NC, USA) was used for statistical analysis by one-way ANOVA. The significance method was LSD (least significant difference), and the detection limit was 0.05.

## 3. Results

### 3.1. Muscle Glycogen

As shown in Table 1, the muscle glycogen content of blunt snout bream decreased considerably as respite time increased (*p* < 0.05). Without respite (0 h), the glycogen content of blunt snout bream muscle was 0.62 mg/g. The increase in the respite time to 24 h resulted in a significant decrease in muscle glycogen to 0.49 mg/g (*p* < 0.05). When the respite time was extended to 48 h, muscle glycogen decreased to 0.44 mg/g, although there was no significant difference in comparison to 24 h (*p >* 0.05). A 72 h respite time resulted in a significant decrease in muscle glycogen to 0.34 mg/g (*p >* 0.05). At 72 h, muscle glycogen dropped by 45.16% in fish muscle compared to the sample without respite (0 h).

### 3.2. Lactic Acid

The lactic acid content of blunt snout muscle decreased as respite time increased, from 3.77 mg/g protein at 0 h to 2.90 mg/g protein at 48 h, and subsequently increased to 3.20 mg/g protein at 72 h (*p* < 0.05). The content of lactic acid in muscle of blunt snout bream after respite was significantly lower than that without respite (0 h). After 48 h of respite before live transportation, their muscles had the lowest amount of lactic acid, at 2.90 mg/g protein (Table 1).

### 3.3. pH

The muscle pH of blunt snout bream increased significantly from 6.71 to 6.98 as the respite time extended from 0 h to 48 h (*p* < 0.05). The muscle pH of blunt snout bream significantly decreased to 6.85 (*p* < 0.05) after live transportation when the respite time continued to increase to 72 h (Table 1). Muscle pH was significantly higher in blunt snout bream after different times of respite (24, 48, and 72 h) compared to that of sample without respite (0 h) (Table 1).

### 3.4. Water-Holding Capacity

After live transportation, the water-holding capacity of blunt snout bream without respite (0 h) was 71.77%. After 24, 48 and 72 h of respite, the muscle water-holding capacity of blunt snout bream increased to 80.45%, 82.56%, and 79.59%, respectively (*p* < 0.05). The water-holding capacity decreased considerably (*p* < 0.05) after 72 h of respite as compared to 48 h, but was significantly greater than that without respite (0 h) (Figure 1).

### 3.5. Color

Table 2 illustrates the effect of respite time before live transportation on the muscle color of blunt snout bream. The lightness (*L**) and whiteness (*W*) of blunt snout bream muscles did not substantially change as respite time increased (*p >* 0.05). Without respite (0 h), the redness (*a**) and yellowness (*b**) values of blunt snout bream muscles were 0.75 and 1.82, respectively. They were significantly higher than the *a** and *b** values of blunt snout bream muscles after respite (24, 48, and 72 h) before transportation (*p* < 0.05). Furthermore, the *a** and *b** values of blunt snout bream muscle did not change significantly after 24–72 h respite time (Table 2).

### 3.6. Shear Force

As shown in Figure 2, the shear force increased significantly as the respite time increased (*p* < 0.05). The muscle shear force of blunt snout bream without respite (0 h) was 486.64 g, which increased to 538.35 g when the respite time was 24 h. It increased to 595.29 and 759.50 g after 48 and 72 h of respite, respectively. Compared with 0 h, the shear force increased by 56.07% after 72 h of respite (Figure 2).

### 3.7. ATP-Related Compounds

As shown in Table 3, the ATP content of muscle from blunt snout bream without respite (0 h) was found to be 25.81 mg/100 g. However, ATP was not detected in the muscles of blunt snout bream with respite (24, 48, and 72 h), showing that the muscle ATP content was low and almost totally degraded. With increasing respite time, the energy compounds ADP and ATP in muscle decreased significantly, from 41.34 and 10.09 mg/100 g to 29.96 and 0.38 mg/100 g, respectively; the umami compounds IMP decreased significantly from 435.43 to 383.01 mg/100 g, and the bitter compounds HxR and Hx decreased significantly from 25.37 and 4.01 mg/100 g to 13.07 and 2.17 mg/100 g, respectively (*p* < 0.05).

The K value of fish meat was less than 10% after 0, 24, 48 and 72 h of respite. Moreover, the K value of the sample without respite (0 h) was significantly higher than those after 24, 48, and 72 h of respite (Table 3).

### 3.8. Morphological Observation

The cross-sectional cytoarchitecture of blunt snout bream muscles with different respite time is shown in Figure 3. The dorsal muscle cells of blunt snout bream were irregularly polygonal in cross-section, with cells closely adjacent to each other. When the respite time was prolonged from 0 to 48 h, the extracellular space in the muscle of blunt snout bream progressively decreased. At 48 h, the cells were the most intact and full. However, when the time of the respite was increased to 72 h, the densely coupled cells progressively disengaged and the extracellular distance expanded substantially (Figure 3).

## 4. Discussion

During the live transportation, water is polluted due to the physiological and metabolic activities (respiration and excretion) of fish. After the deterioration of water quality, fish undergo an intense stress response, which affects their survival rate and muscle quality. Ammonia nitrogen in transportation waters comes mainly from the fish’s own nitrogen excretion and degradation of nitrogenous organic matter such as feces [21]. When the ammonia nitrogen content of water is too high, nonionic ammonia competes with oxygen for hemoglobin, resulting in fish tissue hypoxia [3]. In addition, it may cause damage to the fish’s liver and renal tissues, resulting in edema, congestion, and inflammation, which may even lead fish to unconsciousness and death [22]; Additionally, it may increase the quantity of reactive oxygen species (ROS), resulting in oxidative stress, impairing muscle cell function of fish, and possibly triggering death [23].

The carbohydrates in the bait provide the major energy for blunt snout bream. During the pre-stop feeding period (respite), blunt snout bream might retain a tiny quantity of undigested leftover bait in their bodies. When bait was depleted, they were unable to obtain carbohydrates. To maintain normal physiological functions, the fish would preferentially use glucose stored in the form of glycogen; when glycogen in the body was depleted, they would use muscle fat; and when severely starved, they would even break down muscle protein [23,24]. Therefore, the muscle glycogen content of blunt snout bream gradually decreased with the increase in respite time, and the sample with respite showed significantly lower glycogen content than the blunt snout bream without respite (Table 1). This result is consistent with Einen et al. [25]. Changes in the amount of ATP and its metabolic products can be indicative of changes in fish muscle energy [26]. ATP is synthesized during phosphagen (ATP-Pcr), glycolysis, aerobic oxidation, etc. [27]. ATP was detected only in the sample without respite, at 25.81 mg/100 g (Table 3). The content of ATP-related compounds, including ADP, AMP, IMP, HxR, and Hx, decreased significantly with the increase in respite time (*p* < 0.05). It was probable that the blunt snout bream could not obtain carbohydrates during the respite to replenish energy. Thereby, the glycogen content of muscles progressively diminished, resulting in an ATP breakdown rate exceeding the synthesis rate [28]. Increased ATP catabolism led to its continual breakdown to ADP, which was then broken down further by myokinase to ATP and AMP. By the action of AMP deaminase, AMP was degraded to IMP and ammonia, and IMP accumulated in muscle. The umami compounds are mainly amino acids, nucleotides, and peptides. The nucleotides of AMP and IMP both are umami compounds that contribute positively to the flavor of fish. However, further degradation of IMP produces bitter compounds such as HxR and Hx.

After being aroused by stressors such as hunger, shock, and crowding, muscular activity (swimming, wrestling, and escape behaviors) of the blunt snout bream increased, which accelerated the consumption of energy compounds in the muscles, and consequently changed the fish’s physiological metabolism. In order to sustain normal physiological function, the fish might activate anaerobic glycolysis and the phosphocreatine pathway to synthesize ATP. Glycogen was degraded in muscle through glycolysis to pyruvate, which was then converted to lactate under anaerobic and lactate dehydrogenase conditions, resulting in lactate accumulation in the muscle. The muscles of the blunt snout bream that had been respited before transportation showed a low lactic acid level (Table 1). This might be because the respite slowed ammonia metabolism and emptied the fish’s excretion, which contributed to alleviate blunt snout bream’s stress during harvesting and transportation. As a result, it attenuated the anaerobic glycolytic response, and decreased lactic acid buildup. However, at 72 h respite time, the ATP-relatives and glycogen levels in fish muscles were low. In this case, the fish was short of energy to maintain life activities, which might weaken the capacity for adapting to the new environment [29]. During live transportation, blunt snout bream with too long a respite time (72 h) were more susceptible to ammonia [9], which exacerbated stress response. The lactic acid content increased while muscle glycogen content decreased, which might be caused by anaerobic metabolism (Table 1). The pH of muscle was closely related with the content of lactic acid, i.e., the accumulation of lactic acid led to the decrease in pH. As a result, the pH of blunt snout bream progressively increased as the respite time was prolonged to 48 h. When the respite time continuously rose to 72 h, the muscle pH decreased (Table 1). Additionally, the change in pH might be related to the creatine content. When the respite time was increased to 72 h, the activity of creatine kinase converted phosphocreatine and ADP to ATP and creatine, which may cause the decrease in pH [27].

The pH of muscle has an impact on the net surface charge of myogenic fibronectin. When the pH of muscle is low, the number of net charges on the surface of myogenic fibrous proteins is less, which generally corresponds to a lower water-holding capacity of muscles. The water-holding capacity of the fish muscle is also related to the cell structure, i.e., muscle with intact and homogeneous structure has a high water-holding capacity. After 24–72 h of respite, the cellular structure of blunt snout bream cells was obviously denser than that without respite. However, the cellular structure of blunt snout bream muscle transplanted after 72 h was more porous than that after 48 h of respite (Figure 3). The water-holding capacity of the blunt snout bream’s muscle increased dramatically when the respite time was prolonged from 0 to 48 h. When the respite time was increased to 72 h, the muscle water-holding capacity fell marginally. The changes in cellular structural might be related to the muscle fiber injury and muscular atrophy mediated by the stress response [30,31] and elevated concentrations of reactive oxygen species (ROS). When fish are stressed, the concentration of reactive oxygen species (ROS) rises, prompting an oxidative stress response in the fish. Increased ROS levels may cause lipid peroxidation and the release of intracellular components (lysosomal enzymes) from muscle cells, which damage the cell membrane and cause cells to break apart [32]. Additionally, the ROS interfers with calcium binding and myogenic fibrin breakage in muscle, weakening actomyosin interactions and leading to a decrease in muscle fiber diameter [15].

The shear force of blunt snout bream muscle increased as the respite time increased, which might be mainly associated with the changes in muscle cell structure [33,34]. Excreta such as ammonia excretion and feces from blunt snout bream without respite (0 h) might severely contaminate the water, resulting in increased stress in blunt snout bream. The stressor stimulates receptors in the hypothalamus, resulting in hormonal and electrical signals that activated the hypothalamic–pituitary–adrenal axis, causing it to release glucocorticoids. The nuclear transcription factor NF-κB might be activated as the glucocorticoid production rises, which might result in cell autophagy and death, as evidenced by the ruptured cells [31]. Respite might reduce metabolism rate and oxygen consumption, which leads to lower stress response intensity and weaker cellular damage in the fish during transportation. After the blunt snout bream was subjected to respite treatment before live transportation, the fish might use the fat in the muscle to maintain normal physiological metabolism since it was short on carbohydrate sources. Therefore, significant increases in muscle shear force in blunt snout bream might be a result of muscle lipid depletion as well [35].

The redness and yellowness values of the muscles of blunt snout bream with respite (24, 48, and 72 h) were significantly lower than those without respite (0 h), which might be related to the lower level of stress during live transportation. When fish were stressed, their muscular tissues became congested [36], affecting the color of the muscles. Additionally, as the stress response was intensified, the fish muscle contractions were accelerated, affecting the capacity of astaxanthin and canthaxanthin to bind actinomycin acid [37]. This might change the redness value and the yellowness value of the muscle.

## 5. Conclusions

With respite time at 24 and 48 h, the muscle quality of blunt snout bream obviously improved, but it declined when the respite time was extended to 72 h. The improvement of muscle quality could be mainly attributed to the benefits of bait during the respite (< 48 h), which promoted fish evacuation of digestive tract waste. It might effectively alleviate the fish stress during live transportation. As a result, water-holding capacity and freshness (indicated by K value) increased; shear force increased significantly. However, prolonged respite led to the consumption of stored energy substances (such as glycogen). It might lead to increased stress injury, which could be confirmed by the results of structural changes in muscle cells. Consequently, the water-holding capacity decreased significantly. This research demonstrates that it is suitable to transport live blunt snout bream after 48 h of respite.

## Figures and Tables

**Figure 1 foods-11-02254-f001:**
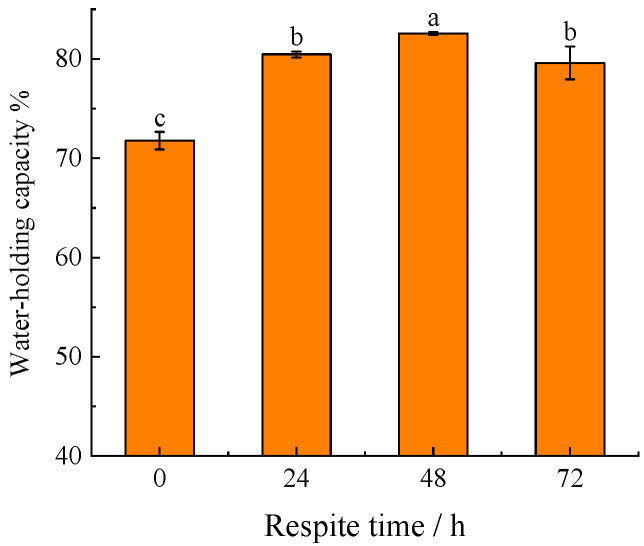
The effect of respite time before live transportation on water-holding capacity of blunt snout bream. Different lowercase letters indicate significant difference (*p* < 0.05).

**Figure 2 foods-11-02254-f002:**
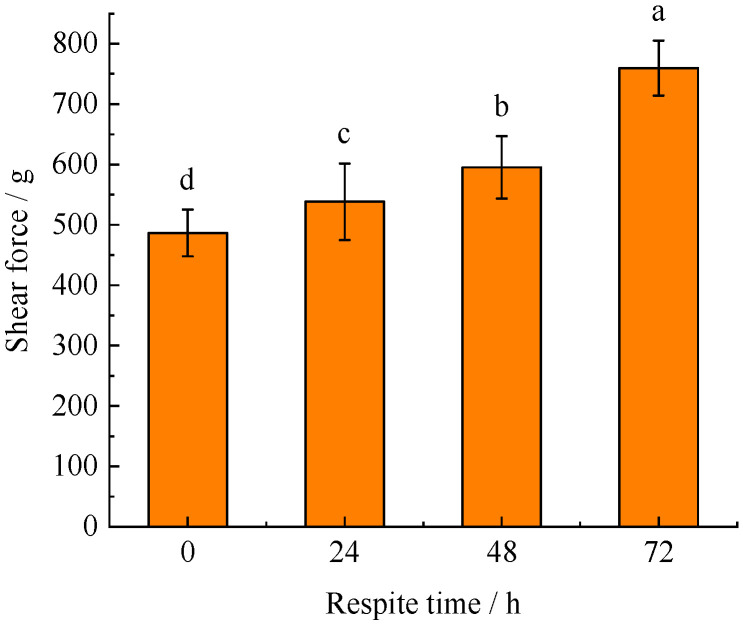
The effect of respite time before live transportation on shear force of blunt snout bream. Different lowercase letters indicate significant difference (*p* < 0.05).

**Figure 3 foods-11-02254-f003:**
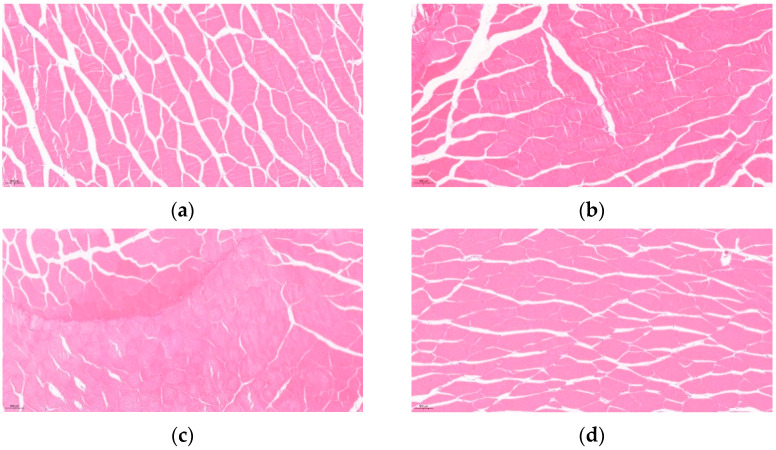
The effect of respite time before live transportation on cross-sectional cytoarchitecture of blunt snout bream: (**a**) 0 h, (**b**) 24 h, (**c**) 48 h, and (**d**) 72 h. Muscle fiber microstructure with 100 times magnification.

**Table 1 foods-11-02254-t001:** The effect of respite time before live transportation on muscle glycogen, lactic acid, and pH of blunt snout bream (n = 3 for each respite time).

Respite Time (h)	Muscle Glycogen (mg/g)	Lactic Acid (mg/g Protein)	pH
0	0.62 ± 0.06 ^a^	3.77 ± 0.17 ^a^	6.71 ± 0.10 ^c^
24	0.49 ± 0.00 ^b^	3.22 ± 0.19 ^b^	6.87 ± 0.08 ^b^
48	0.44 ± 0.00 ^b^	2.90 ± 0.17 ^b^	6.98 ± 0.05 ^a^
72	0.34 ± 0.03 ^c^	3.20 ± 0.16 ^b^	6.85 ± 0.05 ^b^

Notes: different lowercase letters in the same column indicate significant differences (*p* < 0.05).

**Table 2 foods-11-02254-t002:** The effect of respite time before live transportation on muscle color of blunt snout bream (n = 6 for each respite time).

Respite Time (h)	*L**	*a**	*b**	*W*	Δ*E*
0	51.18 ± 1.80 ^a^	0.75 ± 0.27 ^a^	1.82 ± 0.88 ^a^	51.13 ± 1.78 ^a^	0.00
24	51.34 ± 1.83 ^a^	−0.34 ± 0.10 ^b^	−0.36 ± 0.15 ^b^	51.33 ± 1.83 ^a^	2.89 ± 0.69
48	49.47 ± 0.69 ^a^	−0.37 ± 0.19 ^b^	−1.06 ± 0.49 ^b^	49.46 ± 0.69 ^a^	3.56 ± 0.68
72	50.95 ± 1.19 ^a^	−0.37 ± 0.19 ^b^	−1.03 ± 0.32 ^b^	50.93 ± 1.19 ^a^	3.26 ± 0.32

Notes: different lowercase letters in the same column indicate significant differences (*p* < 0.05). *W*: whiteness; **Δ***E*: the total color difference; *L**: lightness; *a**: redness; *b**: yellowness.

**Table 3 foods-11-02254-t003:** The effect of respite time before live transportation on ATP-related compounds of blunt snout bream (n = 3 for each respite time).

Respite Time (h)	ATP	ADP	AMP	IMP	HxR	Hx	K Value
0	25.81 ± 0.41	41.34 ± 0.3 ^a^	10.09 ± 0.37 ^a^	435.43 ± 1.58 ^a^	25.37 ± 0.49 ^a^	4.01 ± 0.56 ^a^	5.87 ± 0.13 ^a^
24	-	38.02 ± 2.06 ^b^	8.20 ± 1.00 ^b^	397.76 ± 0.48 ^b^	16.44 ± 0.78 ^b^	3.19 ± 0.30 ^b^	4.20 ± 0.12 ^b^
48	-	35.08 ± 1.55 ^c^	1.95 ± 0.95 ^c^	389.11 ± 0.33 ^c^	14.48 ± 2.52 ^bc^	3.11 ± 0.38 ^c^	3.94 ± 0.64 ^b^
72	-	29.96 ± 0.64 ^d^	0.38 ± 0.21 ^d^	383.01 ± 0.18 ^d^	13.07 ± 1.86 ^c^	2.17 ± 0.06 ^d^	3.51 ± 0.41 ^b^

Notes: different lowercase letters in the same column indicate significant differences (*p* < 0.05). ATP: 5′-adenosine triphosphate; ADP: 5′-adenosine diphosphate; AMP: 5′-adenosine monophosphate; IMP: inosinic acid; HxR: inosine; Hx: hypoxanthine; K: freshness.

## Data Availability

The data presented is contained within the article.

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
