# Peer review of "Effect of Respite Time before Live Transportation on Muscle Quality of Blunt Snout (Wuchang) Bream"

_foods, 2022, doi:10.3390/foods11152254_

Round 1
Reviewer 1 Report
1. Line 2: Scientific name of Wu-chang Bream is needed.
2. Line 15: significantly
3. Line 16: transportation
4. Line 20: remained
5. Line 59: italicize “Megalobrama amblycephala”
6. Line 90-91. How to maintain the water temperature and dissolved oxygen? Please describe.
7. Line 96-97. Describe the principle of lactic acid and glycogen determinations using kits.
8. Line 102. Why only dorsal was measured for pH? Why not other muscles like dorsal, ventral, lateral line,?
9. Table 2. Please remark the abbreviation used.
10. Table 3. Why ATP was not detected after 24 h? Since glycogen was still remained at that time.
11. The K value can be calculated in Table 3.
12. How about the ammonia N in fish muscle?
13. Line 244-245/307-313. Authors mentioned ROS. So, how about the lipid oxidation and myoglobin redox instability of fish muscle?
14. Fig. 1 and Fig. 2 please discuss about the relationship between WHC, pH, and shear force.
15. Please italicize L*, a*, and b* throughout the manuscript.
16. Line 266-267. Not only IMP is responsible for the umami but also some free amino acids, and peptides. Please elaborate on this.
Author Response
Comments #1
- Line 2: Scientific name of Wu-c hang Bream is needed.
Author: Thanks for your suggestion. It was changed to “blunt snout bream”.
- Line 15: significantly
Author: Thanks for your suggestion. It was corrected.
- Line 16: transportation
Author: Thanks for your suggestion. It was corrected.
- Line 20: remained
Author: Thanks for your suggestion. It was corrected.
- Line 59: italicize “Megalobrama amblycephala”
Author: Thanks for your suggestion. It was corrected.
- Line 90-91. How to maintain the water temperature and dissolved oxygen? Please describe.
Author: Thanks a lot for your question. We control the temperature during transportation by adding ice bag. Two ice bags(400 / per)were placed in each box to maintain the temperature about 10 ℃. The thawed ice bags were replaced by a 400 g ice pack every 1 h until the end of the transportation assay. We control the dissolved oxygen during transportation by control the flow of oxygen.
- Line 96-97. Describe the principle of lactic acid and glycogen determinations using kits.
Author: Thanks a lot for your suggestion.
The principle of lactic acid kits: Using NAD+ as the hydrogen acceptor, LDH catalyzes the dehydrogenation of lactic acid (Lactic Acid) to produce pyruvate, which converts NAD+ to NADH. where PMS delivers hydrogen to reduce NBT to a purple colorant, and the absorbance of the colorant is linearly related to the lactic acid content at 530 nm.
The principle of glycogen kits: Glycogen is dehydrated in the presence of concentrated sulfuric acid to form glyoxal derivatives, which then interact with anthrone to form a blue compound that is quantified colorimetrically with standard glucose solutions treated with the same method. Glycogen is very stable in concentrated alkali solution, so the tissue is heated in concentrated alkali before color development to destroy other components while retaining glycogen.
- Line 102. Why only dorsal was measured for pH? Why not other muscles like dorsal, ventral, lateral line?
Author: Thanks a lot for your question. Different sampling parts will also have an impact on the test results, so the consistency of the sampling parts should be maintained. Moreover, the dorsal muscles accounted for the largest proportion of the fish body and were suitable for the determination of indicators such as color and shear force. In order to maintain the consistency of the experiment, we also measured pH in the back muscle.
- Table 2. Please remark the abbreviation used.
Author: Thanks for your suggestion. It was added.
- Table 3. Why ATP was not detected after 24 h? Since glycogen was still remained at that time.
Author: Thanks a lot for your question. It was discussed in the article (line 286-289).
- The K value can be calculated in Table 3.
Author: Thanks for your suggestion. It was added in Table 3.
- How about the ammonia N in fish muscle?
Author: Thanks a lot for your question. The main objective of this paper is to investigate the effect of respite time on the muscle quality of Wuchang bream after transportation, so we did not test the ammonia nitrogen content in fish meat. We sincerely appreciate your understanding.
- Line 244-245/307-313. Authors mentioned ROS. So, how about the lipid oxidation and myoglobin redox instability of fish muscle?
Author: Thanks a lot for your question. The main objective of this paper is to investigate the effect of respite time on the muscle quality of Wuchang bream after transportation. Through literature review, we found that these changes in muscle quality may be related to the production of ROS and lipid oxidation and myoglobin redox installation of fish muscle. This is need to be further studied. We sincerely appreciate your understanding.
- Fig. 1 and Fig. 2 please discuss about the relationship between WHC, pH, and shear force.
Author: Thanks a lot for your question. The relationship between WHC, pH, and shear force were discussed in line321-325/ 340-341. (The pH of the muscle has an impact on the net surface charge of myogenic fibronectin. When the pH of muscle is low, the amount of net charges on the surface of myogenic fibrous proteins diminishes, and the muscle's capacity to store water is diminished. The decrease of muscle pH will cause the decrease of shear force.)
- Please italicize L*, a*, and b* throughout the manuscript.
Author: Thanks for your suggestion. It was corrected.
- Line 266-267. Not only IMP is responsible for the umami but also some free amino acids, and peptides. Please elaborate on this.
Author: Thanks for your suggestion. We respectfully agree with your comments, we elaborate this in line 292-294.

Reviewer 2 Report
The manuscript is about effect of respite time on muscle quality of wu-chang bream after Live transportation.
Abstract
Erase line 12…. were applied and the authors must add the objective of research was explored the effect of...
Abstract
Erase line 12…. were applied and the authors must add the objective of research was explored
After line 172 add the table 1, line 178 figure 1, line 187 table 2, line 193 figure 2, line 293 table 3 and line 214 figure 3.
Figure 1 and figure 2, they should be centred.
Author Response
Comments #2
Comments and Suggestions for Authors
The manuscript is about effect of respite time on muscle quality of wu-chang bream after Live transportation.
Abstract
Erase line 12…. were applied and the authors must add the objective of research was explored the effect of...
Author: Thanks for your suggestion. It was added.
Abstract
Erase line 12…. were applied and the authors must add the objective of research was explored
Author: Thanks for your suggestion. It was added.
After line 172 add the table 1, line 178 figure 1, line 187 table 2, line 193 figure 2, line 293 table 3 and line 214 figure 3.
Author: Thanks for your suggestion. It was corrected.
Figure 1 and figure 2, they should be centred.
Author: Thanks for your suggestion. It was corrected.

Reviewer 3 Report
Dear Authors,
The presented for evaluation paper is very interesting, but in my opinion some corrections and additions are necessary, which can be used to make the manuscript more readable.
Specific comments:
-please take into consider changing the title, which is currently, in my opinion, a little bit confusing, to: "Effect of respite time before live transportation on muscle quality of Wuchang bream"
- Do I understand correctly that the experiment was carried out on a total of 40 fish? Please clarify, e.g. in table 1, (n = 10 for each respite time)
- please briefly describe in chapter 2.2 model of transporting Wuchang bream in China, especially add information about the transport time in simulated transport platform in this study
- how many repetitions of the color measurement were carried out?
- please change lines 115-117: "a * value is positive indicates the sample is red ....", into: "a positive a * represents red, and a negative a * represents green, a positive b * represents yellow , and a negative b * represents blue "(regard to AMSA Meat Color Measurement Guidelines, 2012)
- Line 191, should be probably: "increased" instead of "decreased"
- please consider changing the layout of tables 1, 2 and 3 (maybe combine them into one common table), i.e. changing respite time and tested traits, so that the means are compared in rows and not as now in columns - I leave the decision to the authors.
Best regards
Author Response
Comments #3
Dear Authors,
The presented for evaluation paper is very interesting, but in my opinion some corrections and additions are necessary, which can be used to make the manuscript more readable.
Specific comments:
-please take into consider changing the title, which is currently, in my opinion, a little bit confusing, to: "Effect of respite time before live transportation on muscle quality of Wuchang bream"
Author: Thanks for your suggestion. It was corrected.
- Do I understand correctly that the experiment was carried out on a total of 40 fish? Please clarify, e.g. in table 1, (n = 10 for each respite time)
Author: Thanks for your suggestion. It was corrected.
- please briefly describe in chapter 2.2 model of transporting Wuchang bream in China, especially add information about the transport time in simulated transport platform in this study
Author: Thanks for your suggestion. The model is respite → capture → transportation→dining table. The transportation time (2-12h) is usually determined by the transportation distance.
- how many repetitions of the color measurement were carried out?
Author: Thanks a lot for your question. 6 repetitions of the color measurement were carried out.
- please change lines 115-117: "a * value is positive indicates the sample is red ....", into: "a positive a * represents red, and a negative a * represents green, a positive b * represents yellow, and a negative b * represents blue "(regard to AMSA Meat Color Measurement Guidelines, 2012)
Author: Thanks for your suggestion. It was corrected.
- Line 191, should be probably: "increased" instead of "decreased"
Author: Thanks for your suggestion. It was corrected.
- please consider changing the layout of tables 1, 2 and 3 (maybe combine them into one common table), i.e. changing respite time and tested traits, so that the means are compared in rows and not as now in columns - I leave the decision to the authors.
Author: Thanks for your suggestion. Tables 1, 2 and 3 represent different aspects of muscle quality, and there is less connection between them. If combine them into one common table, may causing the table to be long and hard to compare. We sincerely appreciate your understanding.

Reviewer 4 Report
Totally, the research has been well designed. Although, there are some points to review.
- Proof editing of English language and style required for all the text.
- The figures and table should be arranged according to text content
- Line 148: please add One-Way ANOVA
Author Response
Comments #4
Comments and Suggestions for Authors
Totally, the research has been well designed. Although, there are some points to review.
- Proof editing of English language and style required for all the text.
Author: Thanks for your suggestion. The language was carefully polished.
- The figures and table should be arranged according to text content
Author: Thanks for your suggestion. It was corrected.
- Line 148: please add One-Way ANOVA
Author: Thanks for your suggestion. It was added.

Reviewer 5 Report
Dear Authors,
The topic of the manuscript entitled ”Effect of Respite Time on Muscle Quality of Wu-chang Bream after Live Transportation” is interesting” but a major revision must be conducted.
comments:
Satistical analysis – Please, explain a numer of repatitions forall analysis (technical and biological repetition).
Please, explain the abbreviations under the Table 2 and under the Table 3.
What about a sensory analysis or instrumental analysis of flavor (GC/MS or electronic nose). Flavor of fish is a very important indicator of its freshness.

Author Response
Comments #5
Dear Authors,
The topic of the manuscript entitled” Effect of Respite Time on Muscle Quality of Wu-chang Bream after Live Transportation” is interesting” but a major revision must be conducted.
comments:
Satistical analysis – Please, explain a numer of repatitions forall analysis (technical and biological repetition).
Author: Thanks for your suggestion. We added the number of repetitions for all analysis after each experimental method.
Please, explain the abbreviations under the Table 2 and under the Table 3.
Author: Thanks for your suggestion. It was added.
What about a sensory analysis or instrumental analysis of flavor (GC/MS or electronic nose)? Flavor of fish is a very important indicator of its freshness.
Author: We fully agree with your comments. In this study, we only detected the flavor compounds of nucleotide. The evaluation indexes of muscle quality mainly include nutritional value, tissue structure, flavor characteristics, sensory quality and physical properties. The color, hardness, water holding capacity and K value can characterize the basic quality of fish. It is also the main evaluation aspect for consumers to buy live fish. Therefore, we did not conduct a comprehensive and systematic study on flavor. we sincerely appreciate your understanding.

Round 2
Reviewer 1 Report
All points raised by reviewers were carefully addressed and answered point-by-point. So, it can be accepted.